# Is the Jaw Bone Micro-Structure Altered in Response to Osteoporosis and Bisphosphonate Treatment? A Micro-CT Analysis

**DOI:** 10.3390/ijms22126559

**Published:** 2021-06-18

**Authors:** Marissa Chatterjee, Fernanda Faot, Cassia Correa, Jente Kerckhofs, Katleen Vandamme

**Affiliations:** 1Department of Oral Health Sciences & Restorative Dentistry, KU Leuven & UZ Leuven, 3000 Leuven, Belgium; marissachatterjee@gmail.com (M.C.); fernanda.faot@gmail.com (F.F.); cassiabcorrea@hotmail.com (C.C.); jente.kerckhofs@uzleuven.be (J.K.); 2School of Dentistry, Federal University of Pelotas, Pelotas 96010-610, RS, Brazil; 3UNICAMP/Piracicaba Dental School, University of Campinas, Piracicaba 13414-903, SP, Brazil

**Keywords:** rat, jaw bone, ovariectomy, bisphosphonates, trabecular bone micro-architecture

## Abstract

The aim of the study was to quantify the micro-architectural changes of the jaw bone in response to ovariectomy, exposed or not to bisphosphonate treatment. A total of 47 Wistar rats were ovariectomized (OVX) or sham-operated (shOVX) and exposed to osteoporosis preventive treatment for eight weeks either with bisphosphonates (alendronate, ALN; group OVX-ALN) three days/week at a dose of 2 mg/kg or with saline solution (untreated control condition; group OVX). The bone morphometric parameters of the trabecular jaw bone were assessed using ex vivo micro-computed tomography. The regions of interest investigated in the maxilla were the inter-radicular septum of the second molar and the tuber. The regions quantified in the mandible included the three molar regions and the condyle. A one-way analysis of variance followed by pairwise comparison using Tukey’s HSD and the Games–Howell test was conducted to explore significant differences between the groups. In the maxilla, OVX decreased the bone volume in the inter-radicular septum of the second molar. Bisphosphonate treatment was able to prevent this deterioration of the jaw bone. The other investigated maxillary regions were not affected by (un)treated ovariectomy. In the mandible, OVX had a significant negative impact on the jaw bone in the buccal region of the first molar and the inter-radicular region of the third molar. Treatment with ALN was able to prevent this jaw bone loss. At the condyle site, OVX significantly deteriorated the trabecular connectivity and shape, whereas preventive bisphosphonate treatment showed a positive effect on this trabecular bone region. No significant results between the groups were observed for the remaining regions of interest. In summary, our results showed that the effects of ovariectomy-induced osteoporosis are manifested at selected jaw bone regions and that bisphosphonate treatment is capable to prevent these oral bone changes.

## 1. Introduction

Oral health is an important component of a person’s general health and quality of life [1]. Elderly patients are now increasingly likely to receive dental treatment for periodontal and orthodontic problems and for replacing missing teeth by oral implants [2]. It is evident that all these treatments target the alveolar bone. Hence the effect of ageing on the alveolar bone is gaining importance with the growing elderly population, especially since the increase in age is associated with an increase in the prevalence of systemic diseases such as diabetes and osteoporosis.

Osteoporotic bone loss has been known to occur commonly in the tibia, femur, and vertebra, whereas the effect of the osteoporotic status on skull vaults is still unclear [3,4,5,6,7,8]. The clinical study of Alam and co-workers on women diagnosed with postmenopausal osteoporosis revealed a divergent radiographic image as an early prediction of osteoporosis [9]. Other clinical studies indicate that osteoporotic women have less mandibular bone mass and bone density compared to their healthy counterparts [10,11]. Animal studies have also shown the link between systemic osteoporosis and bone loss in the jaw, however, these are contentious as the findings varied due to the various assessment methods used and the considered regions of interest [10,12,13,14,15,16,17,18,19,20,21,22]. Among these studies, several found that ovariectomy results in greater bone resorption in the mandibular alveolar bone compared to the maxilla [23,24,25]. Other reports suggest that ovariectomy induced an accelerated mandibular bone loss compared to healthy ageing and that the induced bone loss reached a more moderate, though still increased level over an extended period of time [26,27]. In contrast, Moriya and co-workers stated that ovariectomy itself may not be capable of causing bone loss in the alveolar region [28].

Anti-resorption drugs, such as bisphosphonates, selective estrogen receptor modulators, and hormone therapy have been used to treat osteoporosis for years. These agents suppress bone resorption and bone turnover, resulting in a relative increase in bone mineral density and the subsequent lowering of fracture rates in hip and spine [29,30,31]. The effect of drug treatment for osteoporosis on the jaw bone has been investigated only to a limited extent to the authors’ knowledge [32,33,34,35]. Wang and co-workers noticed a preventive effect of tanshinone, an anti-inflammatory drug, on the jaw bone by suppressing the bone turnover [33]. Icariin, a type of flavonoid, achieved the same outcome though by stimulating the bone formation, as shown in the study led by Xu et al. [32]. In the study of Romuado et al., it was shown that the micro-architecture of the maxilla was not affected by the ovariectomy, nor by the two tested antiresorptive drugs alendronate and odanacatib. Nevertheless, the bone mineral density (BMD) did decrease as a consequence of the ovariectomy. Both drugs could counteract this effect, however only in the case of alendronate the values no longer significantly differed from the control group [34]. Hence the association between estrogen deficiency and bone loss in the jaw remains ambiguous and further research is needed to confirm or reject the influence of estrogen deficiency and its treatment by means of bisphosphonates on jaw bone structural changes. 

The aim of the present study was to investigate the micro-structural characteristics of the jaw bone prone to osteoporosis, in the presence or absence of pharmacological treatment by the bisphosphonate drug alendronate (ALN).

## 2. Results

### 2.1. Maxilla

An increase in bone volume (BV/TV) in the inter-radicular septum of the second molar (M2) was observed for maxillary bone under ALN treatment compared to ovariectomy surgery (OVX) (+11.49%) and sham-ovariectomy surgery (shOVX) (+9.02%). In the maxillary tuber region, there were no statistical differences observed between the three hormonal groups for the quantified trabecular bone parameters (Figure 1).

### 2.2. Mandible

In the region of the first molar (M1), the results of the ellipsoid region of interest (ROI) at the buccal side revealed that ovariectomy caused a marked deterioration of the trabecular bone mass and micro-structure, manifested as a significant decrease in BV/TV (−24.44%) and increase in trabecular spacing (Tb.Sp) (+20.00%) compared to shOVX groups. Furthermore, ALN exerted a positive effect on the number of trabeculae (Th.N) showed by an increase of 19.42% compared with OVX group (Table 1). No significant differences could be observed between all groups in the other ROIs for M1.

For M2, both ROIs (manual drawing/standard ellipse) did not result in any statistical differences related to the bone micro-structure between the three hormonal groups (Table 1).

In the inter-radicular region of the third molar (M3), a negative effect of ovariectomy on the bone micro-architecture was observed with a significant increase in trabecular pattern factor (Tb.Pf) (+185.59%) and structure model index (SMI) (+243.75%) compared to the shOVX group (Table 1).

Finally, for the mandibular condyle head, a significant effect of ovariectomy was detected via an increase in Tb.Pf (+47.40%) and in SMI (+66.66%) compared to the shOVX group. ALN treatment resulted in a significant decrease in Tb.Sp, Tb.Pf, and SMI, and an increase in Tb.N compared to both the shOVX and OVX groups. At the same time, a significant decrease in BV/TV and trabecular thickness (Tb.Th) and increase in total porosity (Tot.Po) were observed in the ALN-treated animals compared to both other conditions (Figure 2).

## 3. Discussion

Trabecular bone is the spongy bone tissue which envelops the roots of the teeth. It plays a vital role in supporting the teeth in the jaw and is often the focus of therapies involving implant restorations, periodontal treatment, and orthodontic movement. With the increasing number of elderly patients and the high prevalence of osteoporosis, trabecular bone has become a prime focus in dentistry. Previous studies showed that after animals had been ovariectomized, their BMD correlated with their jaw bone BMD [27,36,37,38]. Nonetheless, different regions under investigation and unstandardized evaluation methods were adopted in these various studies, resulting in opposing findings [39]. The advent of micro-CT systems made it possible to assess the micro-architecture and structural aspects of the bone in three dimensions. Hence, in the present study, using standardized and optimized measurements for the assessment of the jaw bone micro-structure by means of ex vivo micro-CT [40], the trabecular bone structural changes in the osteoporotic jaw bone, either treated or untreated pharmacologically with bisphosphonates, were assessed. The results showed that the impact of ovariectomy-induced osteoporosis can be seen at the jaw bone level. Moreover, bisphosphonate treatment was able to prevent the maxillary bone loss surrounding the molar tooth. With regard to the mandible, the effect of ovariectomy was seen at M1, M3, and the condyle as shown by a deterioration of the trabecular bone. Treatment with ALN was able to prevent jaw bone deterioration. 

Ever since the U.S. Food and Drug Administration approved the use of ovariectomized rats as a clinical model [41], they have been used in many studies, including alveolar bone studies [37]. Hence ovariectomized rats were used in the current study. As rats achieve sexual maturity at six to eight weeks of age, rats ovariectomized at twelve weeks of age were included. This is in line with the twelve week-old rat model of Lee et al. in their proposed experimental design [20]. Micro-CT as the validated method for assessing the bone micro-architecture [42] was applied to quantify the trabecular bone of the jaw. As there are numerous indicators that can be used to describe the trabecular bone morphology, Bouxsein and coworkers have put forth some guidelines and asserted that at least four of these parameters (i.e., BV/TV, Tb.Th, Tb.Sp, and Tb.N) must be included in the analyses [42,43]. These indicators were included in the present study. However, because the incisors of the rat extend throughout the maxilla and the entire mandible underneath the molars [44], the cortical morphology was difficult to decipher in an accurate and standardized manner and was therefore not included. 

Besides bisphosphonates, denosumab has become a frequently used anti-resorptive drug in the treatment of osteoporosis [45]. Studies comparing denosumab with bisphosphonates suggest that the increase in BMD is more pronounced when denosumab is used [46,47], although a reduction in the fracture risk is not yet proven [48,49]. Among these studies, none have included the jawbones as a region of interest, indicating that the effect of denosumab to the biological and morphological properties of the jaw bones remains unclear and that bone-type-specific studies would be welcomed.

Medication-related osteonecrosis of the jaw (MRONJ) incidence is low (and it is even rarer when antiresorptive medication are used as (preventive) treatment of osteoporosis compared to treatments in oncological settings [50]. Indeed, the pathogenesis of MRONJ involves a combined effect between local infection or trauma and decreased bone turnover after exposure to bisphosphonates or denosumab. The importance of localized dental and periodontal infection in the development of MRONJ has been highlighted recently [50,51,52,53]. Given the fact that the current experiment did not include oral trauma or infection, MRONJ was not expected and not further explored.

The volume of bone in the inter-radicular septum of the maxillary M2 was found to be significantly reduced in OVX rats, suggesting that ovariectomy can cause deterioration of trabecular bone mass just as it does in the long bone and vertebrae [54,55,56]. These results were similar to those observed by Dai et al. [39]. Moreover, an increase in bone volume was seen for the OVX-ALN rats compared to both the shOVX and OVX rats, indicating that ALN has a positive influence on the trabecular bone mass of the maxilla and is able to prevent decrease in bone mass caused by ovariectomy. No significant results were observed in the maxillary tuber.

Amongst the various studies on the trabecular bone of ovariectomized rats in the mandible, Irie et al. [38] observed that, compared to healthy rats, the amount of bone and the Tb.Th in the mandibular M1 showed a decrease of 15% and 16% respectively and the Tb.Sp increased by 18%. In the present study, the region near to the buccal surface of M1 showed a decrease in the BV/TV of OVX rats by 24.43% which surpasses the one showed by Irie et al. Additionally, Tanaka et al. [27] assessed the long-term effect of ovariectomy on the trabecular bone of the mandible and found that BV/TV, Tb.Th, and Tb.N of the OVX rats showed a marked decrease by 75%, 46%, and 58% respectively with a concomitant increase of Tb.Sp by 354% compared to control animals. The latter findings are more striking than the results obtained in the present study by micro-CT evaluation, where Tb.Sp increased by 20%. When comparing the effect of the pharmacological treated group with the OVX rats, the Tb.N increased by 19.42%. The reason for these results could be attributed to the ROI considered in the previous studies as all were in the inter-radicular septum which is a region close to the roots. Since that region is encompassed by the periodontal ligament, the different mastication forces experienced here could lead to different bone remodeling rates [43]. Two studies indeed revealed a positive effect of masticatory forces on the jaw bone by delaying bone loss following ageing and ovariectomy and by improving the overall jaw bone quality [8,57]. Finally, analysis of the bone surrounding the mandibular M3 showed that Tb.Pf and SMI of the ovariectomized rats increased by 185% and 243.75% respectively. As M3 is the smallest tooth among the molars, the quantity of trabecular bone can only be measured in the region of the inter-radicular septum. No other parameters in the other molar regions showed any significant effects. Hence the results of BV/TV, Tb.Sp, Tb.Pf, and SMI indicate a decrease in bone mass and a more porous mandibular region. 

Nevertheless, there are studies where little to no impact on the jaw bones was found after ovariectomy, which is in conflict with our results [58,59,60,61]. Esteves et al. [59] evaluated the effect of the estrogen deficiency on both the maxilla, mandible, and tibia after 60, 90, and 120 days. The tibia showed osteoporotic changes after the 60 day mark, whilst neither jaw bones exhibited any influences after a period of 120 days. In the study of Lee et al. [60], the changes in the rat mandible in response to postmenopausal osteoporosis were examined and compared to those of the femur. A change could be observed in the mandible, however only in some regions and with a delayed and less pronounced response compared to the response of the long bone. 

Besides the molar region, the mandibular condyle also showed significant results with regard to the effect of OVX and ALN administration. This observation closely aligns with the research of Jiang et al. [62], where the ovariectomy-induced bone loss in the condyle was partially inhibited by risedronate treatment. As known, the mandibular condyle is quite distinct as it is one of the regions of the jaw that has multidirectional growth capacity [63,64]. The condyle constantly remains active throughout its lifetime and is quite adaptable to the various functional needs of the mandible [64,65]. The study of Matsumoto et al. assessed the effects of the masticatory activity on the oral condyle in (sh)OVX animals. Their findings suggest that reduction of appropriate mastication activity in the growth period results in poor growth of the mandibular condyle. Furthermore conditions simulating aged/postmenopausal status induce atrophy of the mandibular condyle [57]. In our study, it was observed that OVX caused a significant deterioration of the trabecular bone micro-architecture parameters Tb.Pf and SMI when compared to shOVX. SMI is used to measure the rods and plates in the trabecular bone and gives information about the change in surface curvature that happens when a structure switches between an ideal plate-like to cylindrical or planar shape. In line with the results observed in this study, previous studies also indicate that in OVX rats, there is a trabeculae shift from a plate-like to a rod-like shape resulting in loss of trabecular connectivity and micro-architecture deterioration [66,67]. Furthermore, the treatment with ALN resulted in bone anabolic findings, mainly for Tb.Sp, Tb.N Tb.Pf, and SMI in the condyle. Other trabecular bone parameters though exhibited some negative effects by causing a decrease in BV/TV, Tb.Th, and Tot.Po. The latter findings may be attributed to the effect of bisphosphonate on the chondroclastic resorption. It may be possible that a reduced number of chondroclasts occurred at the chondro-osseous junction due to the ALN treatment. Moreover, ALN tends to accumulate at this junction if the concentration is high because of the level of calcium that usually peaks on day three after administration [68], leading to lowering the activity of these chondroclasts, which ultimately results in failure of mineralization at this osseochondral site [69]. In contrast, the study by Kimura et al. [70], which used a reduced concentration of ALN (1.25/mg/kg/week), did not report this inhibited condylar growth. The findings of the present study suggest that although BV/TV, Tb.Th, and Tot.Po were low, there was no further decrease in bone mass or micro-architecture, relaying that ALN is able to prevent the bone deterioration caused by ovariectomy. This is in line with both the studies of Gomes et al. and Romualdo et al. [34,71]. 

The present study also possesses limitations. It is yet to be seen if the results from the osteoporosis animal experiment can be extrapolated to humans [26]. Furthermore, the ovariectomy model in this study used only one dosage of ALN and one experimental time point. Micro-CT can also only be used in cases of small animal models. If advancements in dental cone beam computed tomography resolution develop, the current methodology may be adopted and applied to analyze the human jaw bone. Moreover, in vivo scanning which ensures monitoring the bone structural changes in real time was not utilized because of image resolution restrictions. Finally, biomechanical testing of the bone strength, in particular the tooth-free condyle region, could have been performed. It also should be noted that more thorough analytical techniques could provide more and better evidence for the influence of estrogen deficiency-induced osteoporosis. 

## 4. Materials and Methods

### 4.1. Animals and Experimental Design

A total of 47 female adult Wistar rats at twelve weeks of age were used in this study. Of them, 31 rats underwent ovariectomy surgery (OVX) while the remaining 16 rats received sham-ovariectomy surgery (shOVX). The surgeries were performed at Charles River Laboratories (Charles River, L’Arbresle, France). For the rats with sham-surgery, their bilateral ovaries were lifted up and returned to their original position, while for the ovariectomized rats the ovaries were removed. Rats arrived five days post-(sh)OVX surgery, with a body weight of 224.8 ± 15 g and 231.1 ± 15 g for shOVX and OVX rats respectively. The OVX group was further divided into two groups: one untreated group (OVX) and one group treated with the anti-resorptive bisphosphonate drug alendronate sodium trihydrate (ALN) (A4978-100MG, Sigma-Aldrich, Bornem, Belgium) (OVX-ALN). ALN was injected subcutaneously three days/week at a dose of 2 mg/kg body weight, starting five days post-OVX surgery for a duration of eight weeks. Saline administration (0.9% NaCl) was performed according to the same time schedule as the ALN administration for the OVX group and for the shOVX group. Injections were administered till the day of euthanasia. The animals were fed standard laboratory diet (or chow) containing 0.7% phosphorus and 1% calcium (SSNIFF, Soest, Germany) and allowed tap water. A pair-feeding regiment was initiated for OVX and OVX-ALN animals at the day of arrival. The average daily food consumption of the shOVX animals was determined and the quantified amount was then provided to all animals in an attempt to control the weight gain for all groups throughout the study. Animals were weighted at the start and once a week during the study. All experiments were conducted according to institutional guidelines for animal welfare; the experimental protocols were approved by the ethical committee of KU Leuven (P130/2010), and followed the ARRIVE guidelines.

### 4.2. Specimen Preparation 

The animals were euthanized by cervical displacement under isoflurane-induced anesthesia. The jaw bones were excised. The maxilla and mandible were retrieved, split at the central axis into two symmetrical parts and immediately fixated in 10% CaCO3–buffered formalin solution (pH 7.4) at 4 °C for 48 h. The samples were further kept in the 70% ethanol at 4 °C until the day of micro-computed tomography scanning.

### 4.3. Micro-Computed Tomography Imaging

An ex vivo desktop micro-computed tomography system, commercially available as Skyscan 1172 (Skyscan, Aartselaar, Belgium), was used for assessment of the micro-architecture of the rat jaw bones [56]. During scanning, the right hemi-maxilla or hemi-mandible were placed in the polyethylene tube and immobilized by means of soft modeling clay. The samples were scanned along the sagittal plane from the condyle towards the molar teeth excluding the incisors for the mandible, and from the dorsal tuberosity up to the molars excluding the incisors for the maxilla. The scanning parameters were 7.8 µm pixel size, 50 kV X-ray voltage, 200 µA electric current, and 0.5 mm Al filter. The dataset was reconstructed with NRecon software (SkyScan, Aartselaar, Belgium). Beam hardening and ring reduction were applied. The reconstructed 3D data sets with a voxel size of 7.8 µm were quantified using CTAn automated image analysis system (Bruker, Kontich, Belgium). The registered gray-value images were segmented into binary images using a Gaussian filter and a fixed threshold (lower and the upper grey threshold values of 70 and 224 respectively) to extract the mineralized bone.

### 4.4. Analysis of the Maxilla

For the maxillary analysis, the ROIs were determined in the inter-radicular region of the distal roots of M2 and in the tuber region. These regions were selected mainly because of ease of visualization of the tooth root when compared to either M1, which consists of five roots, or M3 which has little volume of bone that can be quantified. Furthermore, the tuber is composed mostly of trabecular bone and is a load bearing region due to mastication in the maxilla. 

In order to quantify the trabecular bone in the inter-radicular septum of the M2, a dataset set was created which exclusively contained the M2 region. Using the Data Viewer software (Bruker, Kontich, Belgium), the sample was subsequently re-oriented and the trans-axial dataset was used for analysis as this plane provides optimal visualization of the M2 anatomy. The dataset contained on average 280 to 350 slices. The analysis started by determining a reference point, which was set as the middle slice of the dataset. The drawing of the ROI in the 2D plane was performed on 50 slices (resulting in a 3D volume of interest, VOI) downwards from this reference point. For each slice within the defined VOI, the ROI was delineated manually matching with the area occupied by the trabecular bone (region with porosities and void) by free-hand drawing. The ROIs were interpolated for the entire VOI (Figure 3A).

Likewise, in order to assess the tuber trabecular bone, a subset of images containing exclusively the maxillary tuber region was first created. The total number of slices analysed were set at 50 slices. By using Data Viewer, the sample was reoriented such that the tuber region facing the maxillary sinus was oriented downwards. The trans-axial plane was used for the analysis. With the maxillary tuber being a triangle with the bases facing the maxillary sinus and the vertex facing the buccal cavity, a standard ellipsoid area measuring 0.7 × 0.3 mm^2^ was positioned on each slice at a distance of 1.401 mm vertically from the vertex of the tuber. The ROIs were interpolated for all slices. In this way, the ellipsoid area was delineated and the trabecular bone within this region was quantified (Figure 3B).

### 4.5. Analysis of the Mandible

The ROIs quantified in the mandible included: (i) the bone surrounding the molar region for each individual molar (M1, M2 and M3) and (ii) the mandibular condylar head (Figure 4A). For the bone surrounding the individual molars in the mandible, both semi-automatic and manual delineation of the ROIs was applied. For M1, three ROIs were defined: (i) manually drawn ROI (Figure 4B.1), (ii) standardized ellipsoid ROI containing the trabecular bone close to the lingual surface of the mandible (Figure 4B.2) and (iii) standardized ellipsoid ROI covering the trabecular bone region close to the buccal surface of the mandible (Figure 4B.3). The ROIs for M2 were manually drawn ROI and ellipsoid ROI for the inter-radicular septum (Figure 4B.4). For M3 there was only one ROI selected, namely an ellipsoid ROI for the inter-radicular septum of the roots of the molar. Multiplemandibular ROIs were investigated as it is seen in the literature that when compared to the maxilla the mandible is more susceptible to micro-structural changes in response to ovariectomy in rodents due to its simpler geometry [26,36,72,73,74]. The maxilla on the other hand is characterized by irregular and more complicated roots as well as the presence of the nasal and sinus cavities. By doing so, our approach may also provide insight into which among the three molars experience the most bone micro-structural changes in response to ovariectomy. The final ROI measured in the mandible was the condyle, which is the load bearing region. 

For the actual analysis of the trabecular bone in the mandibular M1 region, as done in the case of the maxilla, a subset of slices containing solely M1 data was created. The dataset contained on average 370 to 400 slices. The reference slice defined was the middle slice of the dataset. The VOI defined for analysis started 100 slices downwards from the reference point and extended over a total of 50 slices. On each of these slices, the ROI was delineated manually by free-hand drawing (Figure 4B.1) as well as via ellipsoid areas with standard dimensions (0.9 × 0.5 mm^2^) positioned at 0.6 mm horizontally from the midpoint of the incisor (Figure 4B.2) and 1 mm vertically from the midpoint of the incisor respectively (Figure 4B.3). Likewise, for the analysis of M2 and M3, a data subset containing solely these teeth was created. The dataset for M2 and for M3 contained on average 218 to 230 slices and 373 to 400 slices respectively. The reference slice was defined as the middle slice of the dataset. The ROI was delineated both manually and by means of an ellipsoid area, over a volume of 50 slices. The ellipsoid area was positioned in between the roots of the molar, such that the edges were not in contact with the periodontal space of the tooth tissues (Figure 4B.4). For M3, the VOI analyzed consisted of 20 slices downwards from the reference slice. For each of these slices, an ellipsoid area measuring 0.4 × 0.4 mm^2^ was positioned in between the roots of M3 as done in the case of M2.

For the condyle analysis, the data subset was reoriented such that the condylar ramus was positioned vertically. The measurements were analyzed on the transaxial images. The VOI consisted of the condylar head extending over 100 slices. Two rectangular ROIs were used within this VOI to assess the trabecular bone micro-architecture, including the porosities of the condylar head. The aim was to include all of the trabecular bone in the condylar head, as the anatomy of the condyle differs between the proximal and the distal regions. The first rectangular ROI with dimensions of 0.5 × 0.4 mm^2^ was projected onto the slices containing the smaller (distal) part of the condyle, while the other rectangular ROI measuring 0.8 × 0.5 mm^2^ was drawn onto the slices with the larger (proximal) condyle part. The ROIs were positioned in vertical direction centrally in the condyle, below the subchondral cartilage (Figure 4A) and were then interpolated for all slices. Investigating the condyle could be interesting as the bone structural changes in the molar region can collectively also alter the load bearing pattern in the mandibular condylar head.

For the trabecular bone in all ROIs, morphometric parameters were assessed following the guidelines put forth by Bouxein and co-workers [42]. The bone volume fraction (BV/TV), trabecular thickness (Tb.Th), trabecular separation (Tb.Sp), and trabecular number (Tb.N) were calculated as 3D measurements of trabecular bone mass and its distribution. Furthermore, the trabecular architecture was evaluated by calculating the connectivity of the trabecular network (trabecular bone pattern factor, Tb.Pf), the structure model index (SMI) and total porosity (Tot.Po). Low Tb.Pf values reflect better connected trabeculae, whereas high Tb.Pf values indicate disconnected trabecular structures. The SMI values provide information regarding the trabecular shape, which is either rod- (SMI = 3) or plate-like (SMI = 0).

### 4.6. Statistical Analysis

Data were expressed as means ± standard error of the means (SEM). Bone morphometrical parameters between the three groups (shOVX; OVX; OVX-ALN) were compared by a one-way analysis of variance (ANOVA) followed by pairwise comparison using Tukey’s HSD and the Games–Howell test, the former being used for equal variance and the latter for unequal variance. The assumption of Gaussian distribution was tested visually using QQ-plots. When indicated, a non-parametric test was performed using the Kruskal–Wallis test with pairwise post-hoc analysis. All the statistical analyses were performed using statistical software (SPSS ver. 22.0, Chicago, IL, USA). Differences were considered significant at *p* < 0.05.

## 5. Conclusions

Based on the experimental setup and the limitations of the study, the conclusions regarding the effect of ovariectomy-induced osteoporosis and its popularly applied pharmacological treatment by means of bisphosphonates in the rodent jaw bone can be summarized as follows:The trabecular bone volume of the maxilla in osteoporotic rats decreases compared to control animals. Bisphosphonate treatment is able to prevent the bone loss due to ovariectomy by increasing the overall bone mass.The bone micro-architecture of the maxillary tuber is not affected in case of ovariectomy-induced osteoporosis.Deterioration of the trabecular bone morphology in response to ovariectomy can also been discerned in the mandible, in particular in the buccal area of M1 and surrounding M3.The mandibular condyle micro-structure is definitely affected by both ovariectomy as well as bisphosphonate treatment.

## Figures and Tables

**Figure 1 ijms-22-06559-f001:**
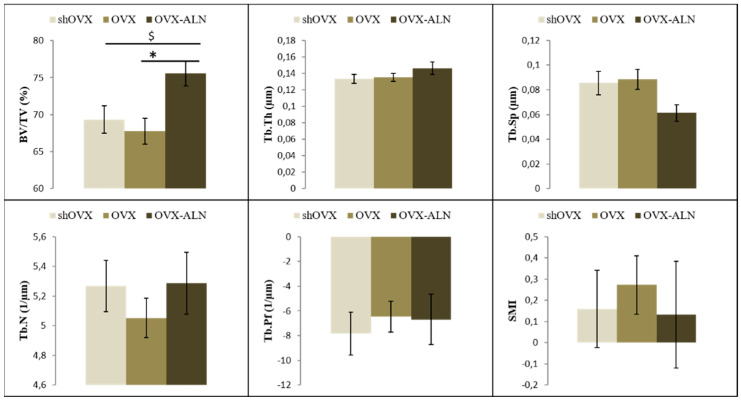
Bar diagrams illustrating the micro-architectural parametric values of the trabecular bone of the second maxillary molar at the inter-radicular septum. *, $: Values with the same sign are statistically significantly different, with *p* < 0.05. BV/TV, bone volume/tissue volume; Tb.Th, trabecular thickness; Tb.Sp, trabecular spacing; Tb.N, trabecular number; Tb.Pf, trabecular pattern factor; SMI, structure model index; shOVX, sham-ovariectomy surgery; OVX, ovariectomy surgery; OVX-ALN, OVX group treated with the anti-resorptive bisphosphonate drug alendronate sodium trihydrate.

**Figure 2 ijms-22-06559-f002:**
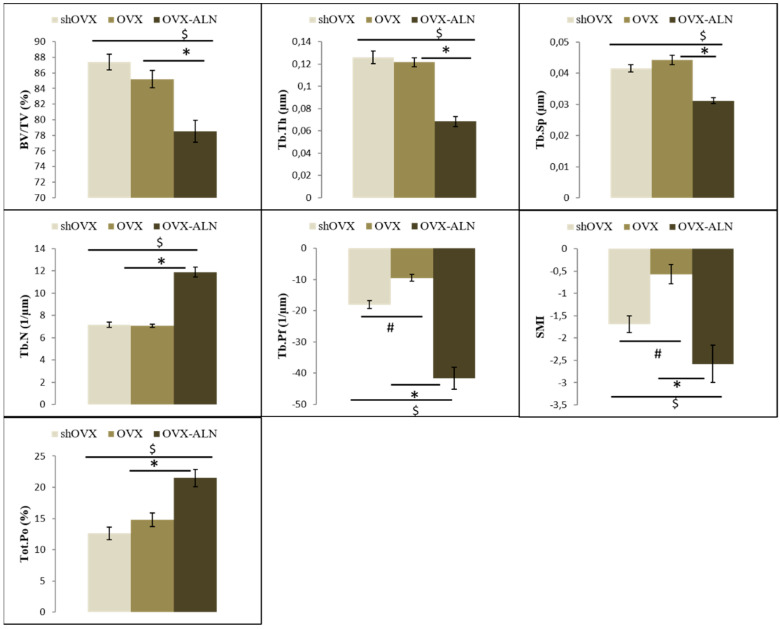
Bar diagrams illustrating the micro-architectural parametric values of the trabecular bone of the mandibular condyle. BV/TV, bone volume/tissue volume; Tb.Th, trabecular thickness; Tb.Sp, trabecular spacing; Tb.N, trabecular number; Tb.Pf, trabecular pattern factor; SMI, structure model index; Tot.Po, total porosity. #, *, $: Values with the same sign are statistically significantly different, with *p* < 0.05.

**Figure 3 ijms-22-06559-f003:**
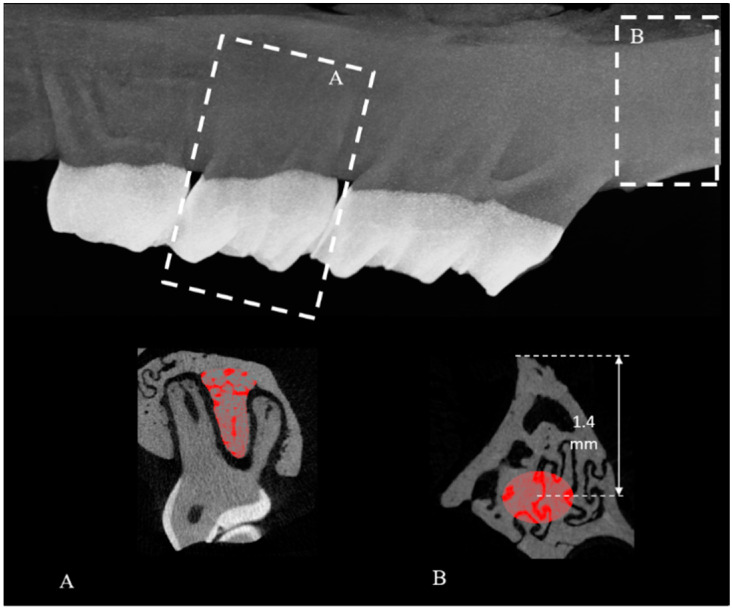
Micro-CT cross-sectional image of the rat maxilla showing the regions of interest (ROI). (**A**) Inter-radicular septum of M2 (manually drawn ROI delineating the trabecular bone); (**B**) Maxillary tuber (ellipsoid-shaped ROI). M: molar tooth.

**Figure 4 ijms-22-06559-f004:**
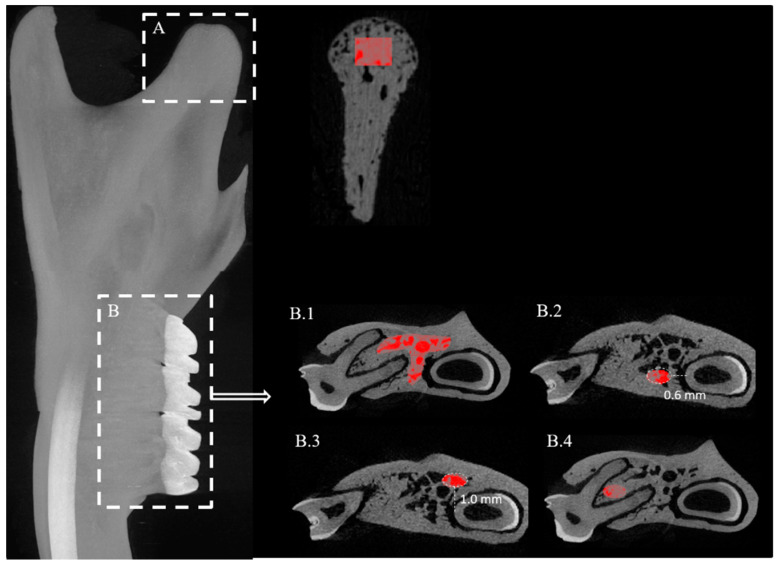
Micro-CT image of the rat mandible showing the regions of interest (ROI). (**A**) Condyle (rectangular-shaped ROI); (**B.1**) Manually drawn ROI encompassing the trabecular bone surrounding the molar; (**B.2**) Ellipsoid-shaped ROI positioned in the lingual region from the roots of the molar; (**B.3**) Ellipsoid-shaped ROI positioned in the buccal region from the molar; and (**B.4**) Trabecular bone in ellipsoid-shaped ROI at the inter-radicular septum.

**Table 1 ijms-22-06559-t001:** Morphometric indices corresponding to the trabecular bone micro-architecture in the molar region of the mandible (*n* = 47).

Parameters	ROI	shOVX (*n* = 16)	OVX (*n* = 15)	OVX-ALN (*n* = 16)
BV/TV	M1	21.41 ± 0.75	20.56 ± 1.00	21.18 ± 0.66
	M1. Lingual	46.87 ± 1.51	44.45 ± 2.54	44.65 ± 2.07
	M1. Buccal	42.76 ± 2.34 ^#^	32.31 ± 2.54 ^#^	37.46 ± 2.5
	M2	18.11 ± 0.73	17.73 ± 0.95	20.73 ± 0.84
	M2. InterRad	82.02 ± 1.90	79.90 ± 2.70	80.12 ± 2.3
	M3. InterRad	81.40 ± 2.46	70.18 ± 5.57	81.37 ± 3.94
Tb.Th	M1	0.11 ± 0.003	0.10 ± 0.003	0.10 ± 0.002
	M1. Lingual	0.14 ± 0.006	0.13 ± 0.006	0.13 ± 0.005
	M1. Buccal	0.14 ± 0.007	0.11 ± 0.007	0.11 ± 0.007
	M2	0.10 ± 0.003	0.11 ± 0.003	0.11 ± 0.003
	M2. InterRad	0.11 ± 0.003	0.11 ± 0.003	0.10 ± 0.004
	M3. InterRad	0.10 ± 0.004	0.09 ± 0.006	0.10 ± 0.004
Tb.Sp	M1	0.30 ± 0.002	0.29 ± 0.004	0.3 ± 0.002
	M1. Lingual	0.15 ± 0.005	0.15 ± 0.007	0.15 ± 0.007
	M1. Buccal	0.15 ± 0.006 *	0.18 ± 0.009 *	0.16 ± 0.009
	M2	0.30 ± 0.005	0.31 ± 0.003	0.32 ± 0.010
	M2. InterRad	0.06 ± 0.005	0.06 ± 0.006	0.06 ± 0.007
	M3. InterRad	0.05 ± 0.003	0.07 ± 0.009	0.06 ± 0.006
Tb.N	M1	1.96 ± 0.06	2.00 ± 0.09	2.13 ± 0.07
	M1. Lingual	3.33 ± 0.11	3.40 ± 0.16	3.30 ± 0.19
	M1. Buccal	3.06 ± 0.08	2.78 ± 0.13 *	3.32 ± 0.19 *
	M2	1.71 ± 0.08	1.6 ± 0.07	1.92 ± 0.12
	M2. InterRad	7.24 ± 0.24	7.36 ± 0.25	7.87 ± 0.36
	M3. InterRad	7.91 ± 0.20	7.02 ± 0.35	7.53 ± 0.25
Tb.Pf	M1	−9.33 ± 1.03	−11.32 ± 1.64	−13.08 ± 2.7
	M1. Lingual	−4.35 ± 0.97	−5.70 ± 2.31	−5.07 ± 2.17
	M1. Buccal	−2.64 ± 1.15	0.36 ± 1.67	−4.20 ± 2.21
	M2	−7.58 ± 1.34	−7.68 ± 1.47	−11.4 ± 2.04
	M2. InterRad	−8.64 ± 2.00	−8.76 ± 2.51	−11.7 ± 3.06
	M3. InterRad	−4.72 ± 1.82 ^#^	4.04 ± 2.81 ^#^	1.03 ± 1.98
SMI	M1	−0.29 ± 0.15	−0.36 ± 0.19	−0.79 ± 0.34
	M1. Lingual	−0.39 ± 0.24	−0.41 ± 0.41	−0.51 ± 0.46
	M1. Buccal	−0.09 ± 0.21	0.44 ± 0.26	−0.34 ± 0.35
	M2	−0.05 ± 0.14	−0.15 ± 0.21	−0.63 ± 0.26
	M2. InterRad	−0.10 ± 0.18	−0.30 ± 0.23	−0.30 ± 0.27
	M3. InterRad	0.32 ± 0.16 ^#^	1.10 ± 0.19 ^#^	0.79 ± 0.17

ROI, region of interest; M1, first molar; M2, second molar; M3, third molar; InterRad, inter-radicular; BV/TV, bone volume/tissue volume; Tb.Th, trabecular thickness; Tb.Sp, trabecular spacing; Tb.N, trabecular number; Tb.Pf, trabecular pattern factor; SMI, structure model index. ^#,^ * Values with the same sign are statistically significant different.

## Data Availability

The datasets generated during and/or analyzed during the current study are available from the corresponding author on request.

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
