# Peer review of "Is the Jaw Bone Micro-Structure Altered in Response to Osteoporosis and Bisphosphonate Treatment? A Micro-CT Analysis"

_ijms, 2021, doi:10.3390/ijms22126559_

Round 1

Reviewer 1 Report

The manuscript submitted to IJMS entitled “Is the jaw bone micro-structure altered in response to osteoporosis and bisphosphonate treatment?” is an original article which aim to evaluate in an animal model the micro-architectural changes of the jaw bone in response to ovariectomy, exposed or not to bisphosphonate treatment.

On my opinion the article is interesting, well written, with good English. Anyway, I tried to split hairs.

Regarding English language minor spell check is required. As for abbreviations, a summary after the conclusion section would be useful. There is no reference to the other drugs used (denosumab 60 mg SC every 6 months) as an alternative to bisphosphonates (especially alendronate) for the treatment of osteoporosis (https://doi.org/10.1056/NEJMoa0809493): please discuss.

A paragraph dedicated to osteonecrosis of the jaws (ONJ), the most fearful complication of pharmacological therapies for osteoporosis such as bisphosphonates and denosumab, is missing (https://doi.org/10.1016/j.joms.2014.04.031). Adequate prevention strategies can avoid the onset of ONJ (https://doi.org/10.1016/j.jcms.2020.01.014).

The other sections of the manuscript were prepared flawlessly.

After making the indicated changes, the article will be suitable for publication.

Thanks for the opportunity to review this manuscript.

Reviewer 2 Report

This is a paper that performed µCT evaluation of the jaw bone for the sham surgery group, ovx group and ovx-bisphosphonates group rats. The effects of ovariectomy extended to the jaw bone, but it was found to be site-specific. It was also shown that these bone loss can be prevented by bisphosphonate treatment.

The following issues should be dealt with prior to acceptance for publication.

You need to choose between table and figure to show the same result.

(All bar graphs are already shown in the previous tables.)

Since it is a result of µCT only, it should be evaluated histologically and bone remodeling, or it should be added to the title that it is a preliminary study.

There is a statistical difference, however what is the clinical difference in the results?

Do you think bisphosphonates should be used to prevent the jawbone loss? I think it is better to add more details about the future prospects including the risk of bisphosphonate-related osteonecrosis.

Author Response

See attachement

Round 2

Reviewer 2 Report

Author’s response has satisfied the requirements of the question of peer review. This manuscript is seems to be acceptable for IJMS.